# Transcriptome Analysis of CYP450 Family Members in *Fritillaria cirrhosa* D. Don and Profiling of Key *CYP450s* Related to Isosteroidal Alkaloid Biosynthesis

**DOI:** 10.3390/genes14010219

**Published:** 2023-01-14

**Authors:** Rui Li, Maotao Xiao, Jian Li, Qi Zhao, Mingcheng Wang, Ziwei Zhu

**Affiliations:** 1College of Food and Biological Engineering, Chengdu University, Chengdu 610106, China; 2Engineering Research Center of Sichuan-Tibet Traditional Medicinal Plant, Chengdu 610106, China; 3School of Basic Medical Sciences, Chengdu University, Chengdu 610106, China; 4Institute for Advanced Study, Chengdu University, Chengdu 610106, China

**Keywords:** *Fritillaria cirrhosa* D. Don, cytochrome P450, isosteroidal alkaloid, full-length transcriptome

## Abstract

*Fritillaria cirrhosa* D. Don (known as Chuan-Bei-Mu in Chinese) can synthesize isosteroidal alkaloids (ISA) with excellent medicinal value, and its bulb has become an indispensable ingredient in many patented drugs. Members of the cytochrome P450 (CYP450) gene superfamily have been shown to play essential roles in regulating steroidal alkaloids biosynthesis. However, little information is available on the P450s in *F. cirrhosa*. Here, we performed full-length transcriptome analysis and discovered 48 *CYP450* genes belonging to 10 clans, 25 families, and 46 subfamilies. By combining phylogenetic trees, gene expression, and key *F. cirrhosa* ISA content analysis, we presumably identify seven *FcCYP* candidate genes, which may be hydroxylases active at the C-22, C-23, or C-26 positions in the late stages of ISA biosynthesis. The transcript expression levels of seven *FcCYP* candidate genes were positively correlated with the accumulation of three major alkaloids in bulbs of different ages. These data suggest that the candidate genes are most likely to be associated with ISA biosynthesis. Finally, the subcellular localization prediction of FcCYPs and transient expression analysis within *Nicotiana benthamiana* showed that the FcCYPs were mainly localized in the chloroplast. This study presents a systematic analysis of the CYP450 gene family in *F. cirrhosa* and provides a foundation for further functional characterization of the *CYPs* involved in ISA biosynthesis.

## 1. Introduction

*F. cirrhosa* D. Don, a member of the Liliaceae family, is a well-known medicinal plant in China. The bulb of this plant, named “Chuan-Bei-Mu” in Chinese, was first introduced in Divine Farmer’s Materia Medica (Shennong Bencao Jing) [1]. It is used as a medicine and healthcare material in many traditional Chinese medicine (TCM) prescriptions [2]. It has been used to treat chronic respiratory disorders such as asthma, cough, lung cancer, and tuberculosis [3]. Modern pharmacological studies have indicated that the bulb of *F. cirrhosa* contained a variety of bioactive isosteroidal alkaloids (ISA) such as imperialine, verticinone, verticine, delavine, peimisine, etc., which are mainly responsible for the anti-asthmatic, antioxidant, antitumor or anti-inflammatory action ascribed to them [4,5,6,7]. Imperialine is a vital ingredient of these *F. cirrhosa* ISA and is frequently utilized as a pharmacopoeia reference standard for the quality assessment of “Chuan-Bei-Mu” due to its significant pharmacological properties [8].

Currently, only extremely low quantities of *F. cirrhosa* isosteroidal alkaloids (FISA), between 0.02% and 0.03% of dry weight bio-mass, can be recovered from “Chuan-Bei-Mu.” [9]. The Fritillaria plants belong to a critically endangered species and are recognized as a Class II protected species in China’s ‘Wild Official Species under Protection of the State according to Wild Medicine Material Protection Rules’ (http://www.cites.org.cn/, accessed on 7 August 2021). Accordingly, further investigation of its pharmaceutical utility is limited by supply shortage [10]. A more promising approach for obtaining structurally complex natural products is the combination of genetic and metabolic engineering [11,12,13,14]. Because of the impact of the environment, Fritillaria is represented in China by various cultivars and variations, each of which has distinct qualities [15]. A multi-species metabolomics investigation revealed that *F. cirrhosa* contains abundant steroidal alkaloids, such as imperialine, suggesting that *F. cirrhosa* D. Don might be considered a suitable plant material for research on FISA biosynthesis [14].

FISAs exhibit a distinctive steroidal architecture with five or six carbocyclic or heterocyclic rings on the C27 steroidal carbon skeleton [1]. FISAs are nitrogen-containing specialized metabolites used as chemophenetic markers in Fritillaria (liliaceae) [16]. In the first step, isosteroidal backbones are produced by the cytosolic mevalonic acid (MVA) and methylerythritol phosphate (MEP) pathways. ISA is then produced via initial cyclization of the central intermediate cycloartenol to cholesterol, followed by hydroxylation, oxidation, and transamination [14]. The majority of the research has been devoted to the cloning and control of functional genes upstream of the terpenoid biosynthesis pathway, including the 3-hydroxy-3-methylglutaryl coenzyme A reductase gene (HMGR), farnesyl diphosphate synthase (FPS), squalene synthase (SS), and cycloartenol synthase (CAS) [15]. Secondary metabolic pathways following cycloartenol generation are rarely studied, particularly the downstream CYP450 genes responsible for C-22, C-23, and C-26 hydroxylation in the steroidal alkaloid biosynthesis pathway.

Here, we used various bioinformatic tools to identify and analyze CYP450 superfamily genes from the full-length transcriptome of *F. cirrhosa*, including gene nomenclature, evolution, structure, and expression patterns. More importantly, we identified candidate CYPs potentially involved in ISA biosynthesis using co-expression analyses and previous reports that were validated by quantitative polymerase chain reaction (qPCR) analyses. Last, the subcellular localization of candidate FcCYP450s were predicted using online tool, two of which were analyzed in *N. benthamiana* leaves. Our study on *CYPs* genes in *F. cirrhosa* provides essential information for further exploration into their physiological and biochemical functions, as well as their application in improving ISA content in Fritillaria plants.

## 2. Materials and Methods

### 2.1. Plant Materials, RNA Extraction, and Sequencing

The plants of *F. cirrhosa* D. Don were harvested inside the wild in July 2020 at Kangding city, northwestern China (placed at 30°3′44.9″ N, 101°58′3.81″ E, altitude 4300 m). The specimen (No. F-200725–1) was deposited at the Engineering Research Center of the Sichuan-Tibet Traditional Medicinal Plant, Chengdu, China. We certify that the research program complies with all applicable institutional, national, and international guidelines and legislation, and that we have authorization to collect *F. cirrhosa.* After collection, the bulbs were rinsed with cold distilled water, sterilized for in vivo culture, frozen in liquid nitrogen, and kept at −80 °C until they were needed again.

An RNA Kit (Vazyme, Nanjing, China) was used to extract the total RNA from the bulbs, and it was then processed in accordance with the manufacturer’s instructions. The purity and concentration of the RNA were determined using a Nanodrop microspectrophotometer (Thermo Scientific, Waltham, DE, USA) and an Agilent 2100 bioanalyzer (Agilent Technologies, Santa Clara, USA). PacBio Iso-Seq library construction followed the official protocol described by Pacific Biosciences (Menlo Park, CA, USA). The PCR products were size selected using BluePippin™ Size Selection System (Sage Science, Beverly, MA, USA), and fragments with 0.5–6 kb were retained. A large-scale PCR was performed to construct the next SMRTbell library. Finally, the PacBio RS II platform was used by Berry Biotechnology Co. (Beijing, China) to complete the sequencing of the four SMRT cells.

### 2.2. Bioinformatics Analysis

For comprehensive functional annotation, the full-length transcripts were compared against public protein databases, including the NCBI nonredundant protein (Nr) database (http://www.ncbi.nlm.nih.gov, accessed on 5 July 2022), Swiss-Prot protein database (https://web.expasy.org/docs/swiss-prot_guideline.html, accessed on 5 July 2022), Kyoto Encyclopedia of Genes and Genomes (KEGG, http://www.genome.jp/kegg/kegg2.html, accessed on 10 July 2022), the COG/KOG database (http://www.ncbi.nlm.nih.gov/COG, accessed on 10 July 2022), and TrEMBL using BLASTx with a cut-off E-value of ≤10–5 (https://blast.ncbi.nlm.nih.gov/; version 2.2.23, accessed on 18 July 2022). Gene Ontology (GO) annotation was analyzed by Blast2GO software (https://www.blast2go.com/; version 4.4, accessed on 18 July 2022) and GO function categories were performed using WEGO software (http://wego.genomics.org.cn/; version 1.0, accessed on 22 July 2022).

### 2.3. Identification, Sequence Analyses and Phylogenetic of FcCYPs

The predicted CYPs were first searched for using keywords against the aforementioned annotations. The National Center for Biotechnology Information (NCBI)’s BLASTx tool was used to manually examine each potential CYPs sequence. After filtering out repeated results, the coding sequences of the resultant subjects were retrieved. ORF Finder (https://www.ncbi.nlm.nih.gov/orffinder/, accessed on 27 July 2022) was used to identify the open reading frame (ORF) sequences of CYPs. The nucleotide sequences of the ORF were translated into amino acid sequences using the ExPASY translation tool (https://web.expasy.org/translate/, accessed on 2 August 2022). The theoretical isoelectric point and molecular weight of each identified CYPs protein were predicted using the ‘‘Compute pI/MW’’ tool in the ExPASy server (https://web.expasy.org/protparam/, accessed on 2 August 2022). The protein subcellular localization of CYPs was predicted using WoLF PSORT (https://www.genscript.com/wolf-psort.html, accessed on 8 August 2022). The names of the CYPs proteins were assigned by Prof. David Nelson [17]. The sequences were divided into A-type, which included only the CYP71 clan, and non-A-type, which included all other clans based on CYP family membership. The CYPs proteins identified here were selected as query sequences to do the BLASTp (https://blast.ncbi.nlm.nih.gov/; version 2.2.23, accessed on 8 August 2022) searches against all available sequences in the Nr database. The full-length sequences of FcCYPs were used to create a phylogenetic tree with other known plant CYPs sequences in public databases using the neighbor-joining (NJ) method with MEGA 11 software at 1000 bootstrap replicates and default settings (Substitution type: Amino acid; Model/Method: Jones-Taylor-Thornton (JTT) model; Rates among Sites: Uniform Rates; Gaps/Missing Data Treatment: Complete deletion).

### 2.4. RNA Extraction and qRT-PCR Analysis

Total RNA was isolated from the bulbs as described above. We examined transcript expression patterns across different tissues (bulb, stem, and leaf). Gene expression levels were determined by qRT-PCR using ChamQ SYBR qPCR Master Mix (Vazyme) on a Bio-Rad CFX96 system (Bio-Rad, Hercules, CA, USA).18S ribosomal RNA was selected as the reference gene. All the primers used are listed in Appendix A. The relative expression was calculated using the 2^−ΔΔCt^ method. The experiment was performed with three biological and technical replicates.

### 2.5. Determination of Alkaloids by HPLC-ELSD

The working standard solutions for the calibration curves were created by diluting the stock solutions of imperialine (420 g/mL), peimine (413 g/mL), and peiminine (406 g/mL) with methanol. The samples were prepared for alkaloid analysis, as described by Ma [18]. The HPLC-ELSD conditions were used to determine the alkaloids, and a liquid chromatograph (Agilent 1260 Infinity, Waldbronn, Germany) equipped with an Alltech 3300 evaporative light-scattering detector was used. The chromatographic separations were carried out a WondaSil C18-WR column (250 mm × 4.6 mm, 5 µm; Agilent) at a column temperature of 30 °C. The column was eluted with a mixture of methanol (mobile phase A) and 0.02% diethylamine (mobile phase B) at a flow rate of 1.0 mL/min. The following elution conditions were used: 0–10 min, 60% A to 75% A; 10–25 min, 75% A to 82% A; and 25–45 min, 82% A to 90% A. The ELSD’s drift tube temperature was set to 60 °C, and nitrogen was used as the carrier gas at a flow rate of 1.2 L/min, gain value of 2, and injection volume of 20 µL. The calibration curves for each compound were used to determine the concentration based on an external standard method.

### 2.6. Subcellular Localization Analysis

We chose the *CYP90A1* and *CYP90B27* genes and used PCR-amplifying the ORFs for these genes without a stop codon while using specific primers with corresponding enzyme sites (Appendix A) to assess representative CYP450 subcellular localization. The isolated PCR products were then subjected to sequence validation by Sangon Biotech (Shanghai, China), after which they were inserted into the pCAMBIA1300-35S-YFP vector upstream of the enhanced yellow fluorescent protein (YFP) in order to produce p35S::CYP450-YFP vectors. These recombinant plasmids were transformed into *N. benthamiana* along with the OsRac3-mCherry plasmid using an *Agrobacterium tumefaciens*-mediated transient transformation system. The plasma membrane localization marker was the fusion protein OsRac3-mCherry as previously described [19]. The cultured *A. tumefaciens* cells (OD600 = 0.4) were infiltrated into the leaves of *N. benthamiana,* which were grown in a greenhouse for 4 weeks at 23 °C under a 16 h light/8 h dark cycle. The same process using the empty vector pCAMBIA1300-35S-YFP was used as the control. Leaves expressing the resultant YFP fusion proteins were visualized 48 h after infiltration using a Nikon Eclipse Ni-U microscope (Nikon, Tokyo, Japan).

## 3. Results

### 3.1. SMRT Sequencing, Similarity Analysis, and Functional Annotation

The sequencing platform PACBIO RS II was used to carry out single-molecule real-time (SMRT) sequencing. A total of 71.35 Gb raw reads were generated and 47,579,643 subreads were obtained after filtering, among which 907,484 reads of inserts (ROIs) were successfully extracted with mean lengths of 1772 bp and 48 passes. In total, 55,751 full-length consensus isoforms, including 55,101 polished high-quality (HQ) and 650 low-quality (LQ) transcripts, were produced.

To obtain a comprehensive annotation of the *F. cirrhosa* transcriptome, a set of 55,101 polished high-quality isoforms was annotated by searching five databases (GO, KOG, Nr, KEGG, and Swiss-Prot). In total, 23,388 isoforms were annotated. Moreover, the specific details of the overall functional annotation are described in Appendix A. In addition, the 70 unannotated isoforms may represent novel *F. cirrhosa* genes.

The gene functions of the identified isoforms were classified using GO enrichment analysis according to their molecular function, cellular component, and biological process terms (Appendix A). Among them, biological processes (30,678 genes) comprised the majority of the GO terms. A significant portion of the isoforms were assigned to GO categories such as cellular processes, metabolic processes, and catalytic activities, which are crucial processes in plants and are involved in metabolite biosynthesis. According to the KOG classification, 14,083 isoforms were identified (Appendix A). The largest group was the cluster for general function prediction (2420).

### 3.2. Identification and Analyses of FcCYPs Sequences

A total of 189 sequences encoding putative CYPs were selected from the *F. cirrhosa* full-length transcriptome by keyword searching and bioinformatics methods, of which 44 lacked the complete signature motifs of P450, and only the longest translation form was retained to present each gene. Eventually, a total of 48 *FcCYP* genes were chosen for the following analyses (Table 1). According to P450 nomenclature, all 48 FcCYP genes were classified into 25 families and 46 subfamilies. They can be divided into two types: A-type (9 families) with 23 genes and non-A type (16 families) with 25 genes. The largest family among them is the CYP71 family, which has 23 members. In contrast, the CYP51, CYP711, CYP95, CYP63, CYP40, and CYP710 families each have just one gene. The deduced FcCYP protein sequences ranged in length from 195 amino acids (CYP728B29) to 689 amino acids (CYP97A10), corresponding to a molecular mass of 21.69 to 76.98 kDa for the proteins.

To explore the evolutionary relationship between FcCYPs, an unrooted NJ tree with 48 full-length FcCYP sequences was built (Figure 1). According to phylogenetic analysis, FcCYPs were classified into 10 clans, including six single-family clans (CYP95, CYP710, CYP711, CYP40, CYP51, and CYP63) and four multi-family clans (CYP71, CYP86, CYP85, and CYP72, in descending order, by number). Genes from the same clan were grouped together to form a single clade. For example, 86 clans, comprising eight CYP450s belonging to five families, were clustered into one clade. The CYP71 clan is regarded a special cluster with the most members.

### 3.3. Identification of Candidates CYPs Involved in ISA Biosynthesis

Four pathways, namely the mevalonate, phytosterol, cholesterol, and core ISA pathways, are involved in ISA biosynthesis [14,15]. We specifically evaluated CYP450s involved in the synthesis of FISA. To date, 47 *CYP450s* have been found to play a role in sterol biosynthesis [20,21,22]. The CYP51G1, CYP94N, and CYP94A subfamilies are the primary CYP450 gene families involved in steroid diversification, with the CYP90 family being the largest multifunctional C-22/23 hydroxylation family involved in steroid biosynthesis [23,24,25]. To identify the most relevant CYP450 unigenes involved in FISA biosynthesis, we performed BLASTp searches that compared FcCYP450s to 51 CYP450s from other plant species known to be involved in steroid metabolism. This analysis showed that CYP90B (contig 25638) exhibited 73.72% sequence identity with *Veratrum californicum* CYP90B27v1, which is a steroid C-22 hydroxylase involved in verazine production [26]. BLASTp analysis also indicated that contig22207 exhibited 63.03% identity with *Oryza sativa* CYP90D2, which catalyzes the C-23 hydroxylation of brassinosteroid [27].

To further investigate the functional roles and evolutionary relationships of these FcCYPs, NJ phylogenetic trees were constructed using 48 CYPs and plants known to be involved in steroid metabolism (Figure 2). Phylogenetic analyses revealed that FcCYP94A-22541 and FcCYP94N-27487 belong to the CYP94 family and are most closely related to *V. californicum* CYP94N1v2, which encodes the C-26 hydroxylation/oxidation enzyme involved in cyclopamine production [26]. Four contigs (CYP72A-18891, CYP72A-23630, CYP72A-24173, and CYP72A-14155) clustered in the CYP72A group and were most closely related to *Solanum lycopersicum* GAMEs [28], *Trigonella foenumgraecum* CYP72A613, and *Paris polyphylla* CYP72A616 [21], which are involved in steroidal glycoalkaloid and diosgenin biosynthesis.

### 3.4. Analysis of Candidate CYPs Gene Expression and Key FISA Accumulation in Bulbs of Different Ages

As the CYP90, CYP72, and CYP94 family genes are involved in steroid metabolism, we examined their expression in relation to alkaloid accumulation in bulbs of various ages. We found that all the selected genes were transcribed more in bulbs from 3 year-old *F. cirrhosa* plants than in bulbs from 1 year-old plants (Figure 3). To verify whether the accumulation of FISA at various ages in *F. cirrhosa* was consistent with the expression levels of these genes, we used HPLC-ELSD to examine the contents of three major FISA (imperialine, peimine, and peininine) in the bulbs of *F. cirrhosa* from the two different age groups (Figure 4 and Appendix A). We found that the accumulation of the three FISA in bulbs cultivated for 3 years was significantly higher than that in bulbs cultivated for 1 year, similar to the trend of the relative expression levels of the candidate CYPs genes. This result suggests that FcCYP proteins are correlated with the FISA biosynthesis.

### 3.5. Assessment of the Subcellular Localization of Two CYP450 Fusion Proteins

Most plant CYP450s target the endoplasmic reticulum (ER) and are occasionally associated with plastids, plasma membranes, and other organelles [29,30]. We used an online tool WoLF PSORT to predict the subcellular localization of 48 FcCYP proteins. Most FcCYP proteins were located in membrane-bound chloroplasts (36), some in the plasma membrane (5), and a few in the cytoplasm (3), nucleus (2), cytoskeleton (1), and vacuolar membrane (1) (Table 1).

To determine the subcellular localization of *F. cirrhosa* CYP450 proteins, we transiently expressed CYP90A1-YFP and CYP90B27-YFP fusion proteins in *N. benthamiana* epidermal cells. OsRac3, a plasma membrane localization marker [19], was also co-transformed with FcCYP fusion proteins. As shown in Figure 5, the fluorescence of CYP90A1-YFP and CYP90B27-YFP partially overlapped with that of OsRac3-mCherry. Meanwhile, the fluorescence of both seemed to be in the chloroplast, some other plastids, and cytoplasm, which was consistent with the Wolfpsort predictions. These results suggest that FcCYP proteins function by targeting different organelles in the cytoplasm.

## 4. Discussion

*F. cirrhosa* is a well-known medicinal herb containing imperialine (a steroidal alkaloid), health products, and landscape, and its bulb is a pharmaceutical and breeding organ. Previous studies on *F. cirrhosa* have mainly focused on cultivation, breeding, phytochemistry, and pharmacology [31]. Because of its highly heterogeneous, low-abundance repeat-derived DNA and large genomes, assembling the genome of *F. cirrhosa* is difficult, time-consuming, and expensive [32]. As a result, little information on the genetic characteristics has been obtained. Although RNA-Seq can be used to explore and analyze differentially expressed genes, the sequences are frequently incomplete. As a result, PacBio Iso-Seq is a more effective and cost-effective method for directly producing a comprehensive transcriptome with precise genetic structural characteristics for *F. cirrhosa*.

In comparison to previous gene discovery studies in *F. cirrhosa* using cDNA cloning or Illumina sequencing [11,13,14], we generated a comprehensive transcriptome dataset. Our Iso-Seq data generated a total of 55,101 deredundant high-quality isoforms, and the full-length transcripts could be directly used for future gene discovery research without additional PCR amplification. Using a combination of SMRT analyses, phylogenetic trees, and gene expression level analyses, we recharacterized the crucial genes encoding CYPs involved in FISA synthesis.

Steroidal alkaloids (SA) are the most common active ingredients in the Liliaceae family, and their biosynthetic pathways of SA have been investigated in many plants, such as *Paris* [24], *Veratrum* [26], and *Fritillaria* species [15]. The CYP51 clan member genes, which descended from a sterol-metabolizing CYP51 ancestor, are thought to be the most ancient CYP450s in the SA biosynthetic pathway. In the present analysis, we identified only one CYP51 clan member (CYP51G1). In *V. californicum* Jervine-type SA synthesis, three specific cytochromes P450 (CYP90B27, CYP94N1, and CYP90G1) exhibit hydroxylation/oxidation activity at the C-22 and C-26 positions. In *F. cirrhosa,* peimisine is a Jervine-type SA, and BLAST and phylogenetic tree analysis showed that FcCYP90B-25638 obtained in this study shares high identity with VcCYP90B27 [26]. CYP90B subfamily members can encode cytochrome P450s that participate in the C-22 hydroxylation of cholesterol [24,26]. In the CYP86 clan, FcCYP94A-22541 and FcCYP94N-27487 are highly similar to VcCYP94N1v2, which encodes steroid C-26 hydroxylase/oxidase. Imperialine is classified as a cevanine-type alkaloid with a ring scaffold typical of cholesterol, and includes a series of hydroxylation/oxidation reactions at C-22 and C-26. Therefore, FcCYP90B-25638, FcCYP94A-22541, and FcCYP94N-27487 were identified as candidate CYPs involved in FISAs biosynthesis.

We then performed a co-expression analysis of key FISAs with gene expression levels in bulbs of *F. cirrhosa* of different ages. We measured the contents of key FISAs in *F. cirrhosa* of various ages, including imperialine, peimine, and peininine. We predicted candidate CYPs that are highly relevant to FISAs. We conducted qRT-PCR to analyze the gene expression levels and found that all the CYPs genes were expressed at higher levels in bulbs in the late growth stage than in the early stage, in line with steroid alkaloid content, which is also in accordance with the levels of alkaloids in the growth stages of *F. taipaiensis* and *F. cirrhosa* [33,34]. This finding suggests that the selected CYPs may be involved in the formation of hydroxylation/oxidase of FISAs. The prediction and analysis of the subcellular localization of FcCYP proteins showed that FcCYP proteins can function in the chloroplast, plasma membrane, endoplasmic reticulum, nucleus, and some plastids. They play key roles in the biosynthetic pathways of SA in different organelles. However, the functions and mechanisms of these conversions remain unclear. We plan to verify these conjectures in *Saccharomyces cerevisiae* heterologous expression in a follow-up study.

## 5. Conclusions

In this study, we identified 48 putative CYP450s with complete cytochrome P450 domains in the full-length transcriptome of *F. cirrhosa*. According to the classification criteria, 48 CYP450s were classified into 10 clans consisting of 25 families and 46 subfamilies. Following that, we conducted phylogenetic analysis to characterize 20 CYP450s identified as being involved in FISAs biosynthesis. The qRT-PCR and HPLC-ELSD results showed that all of the predicted candidate *CYP450s* genes expressed similarly with key FISAs concentrations in bulbs of different growth ages. Finally, the subcellular localizations of the two CYP450 candidates were investigated. Together, this study provides thorough understanding of the CYP450 gene families in *F. cirrhosa* and will help in figuring out how the CYP450 family functions in this and related species. Further research is required to confirm the role of the selected candidate *CYP450s* genes in FISA biosynthesis.

## Figures and Tables

**Figure 1 genes-14-00219-f001:**
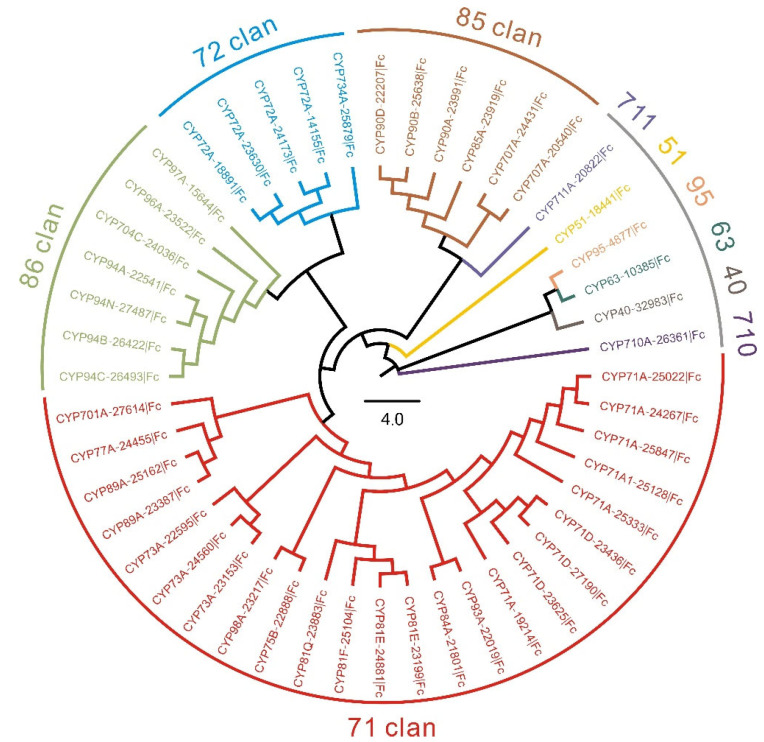
Phylogenetic analysis of predicted CYP450s in *F. cirrhosa*.

**Figure 2 genes-14-00219-f002:**
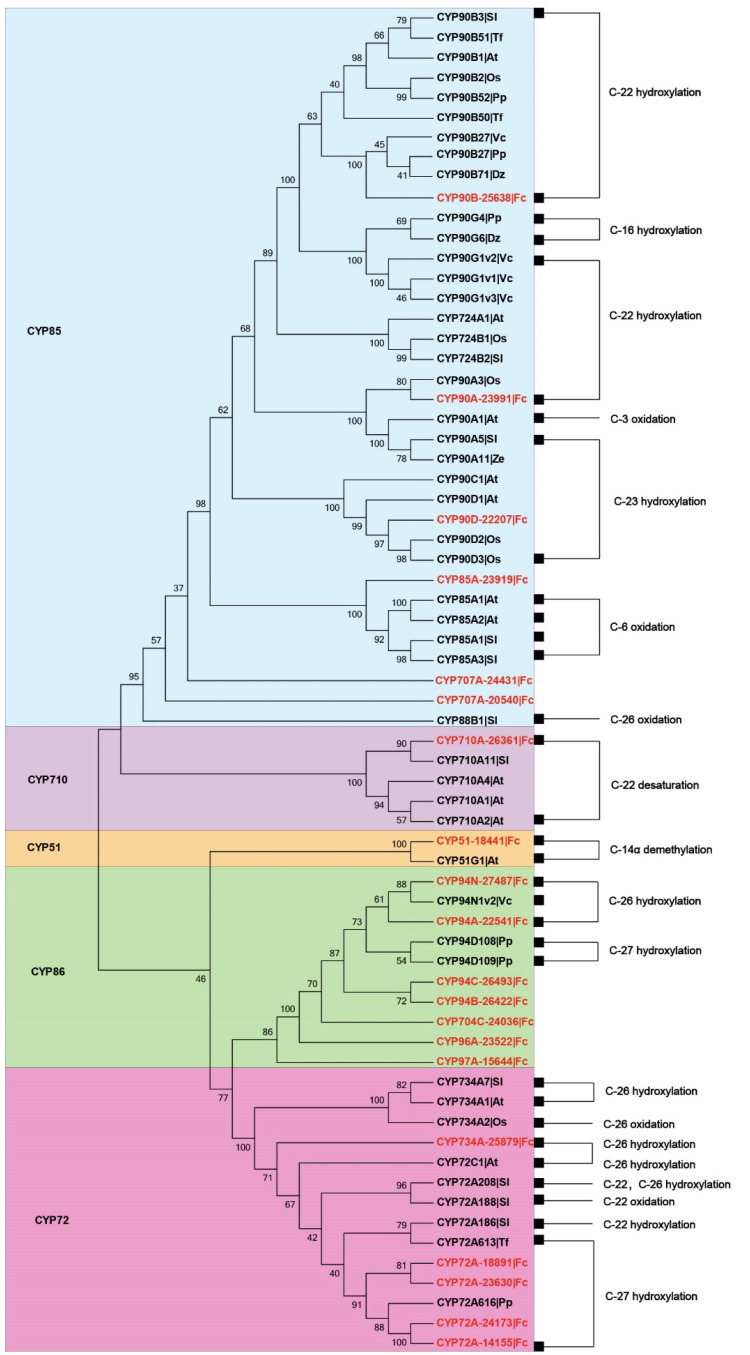
Phylogenetic tree of previously characterized sterol biosynthesis *CYP450s* and those *F. cirrhosa CYP450s* isolated in this study (in red). The known biochemical activities of *P450s* are indicated on the right.

**Figure 3 genes-14-00219-f003:**
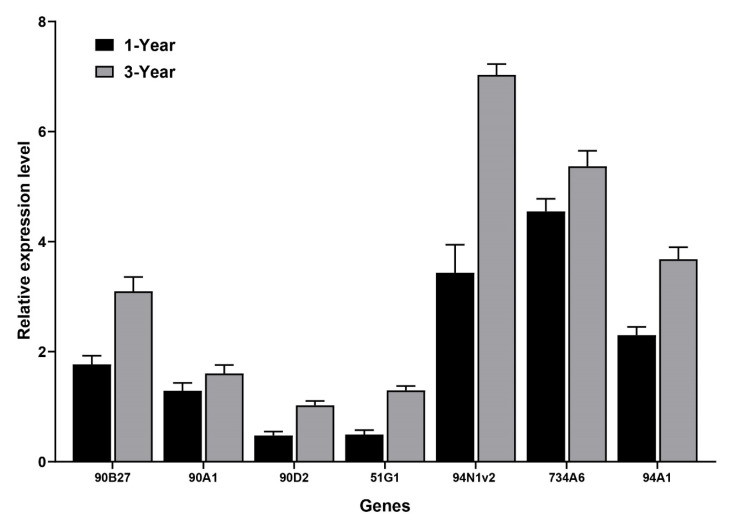
The relative expression levels of candidate *CYPs* in *F. cirrhosa* bulbs of 1 year of age and 3 years of age. The error bars indicate ± SDs (*n* = 3).

**Figure 4 genes-14-00219-f004:**
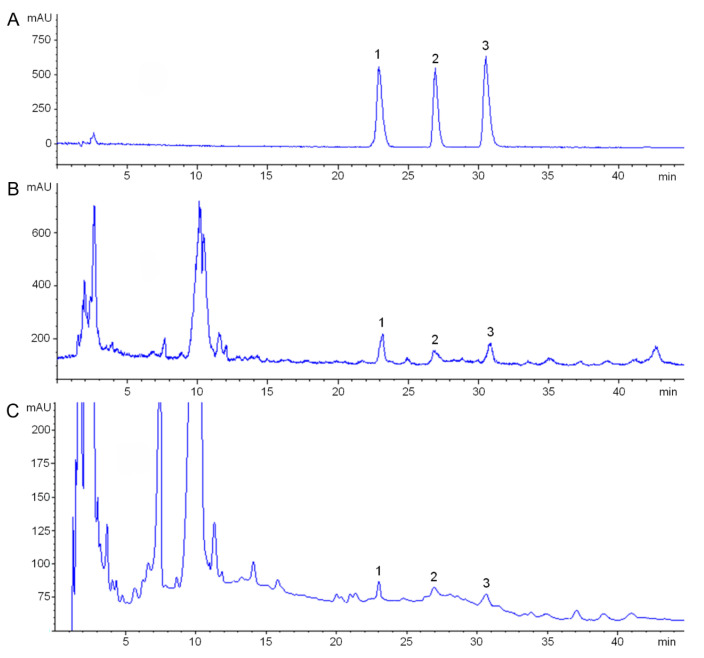
Representative HPLC-ELSD chromatographic profiles of FISA in the bulbs of *F. cirrhosa*. (**A**) HPLC-ELSD chromatographic profiles of ISA standards. (**B**) HPLC-ELSD chromatographic profiles of FISA in the bulbs from 3-year-old *F. cirrhosa*. (**C**) HPLC-ELSD chromatographic profiles of FISA in the bulbs from 1-year-old *F. cirrhosa* bulbs. 1, imperialine; 2, peininine; 3, peimine.

**Figure 5 genes-14-00219-f005:**
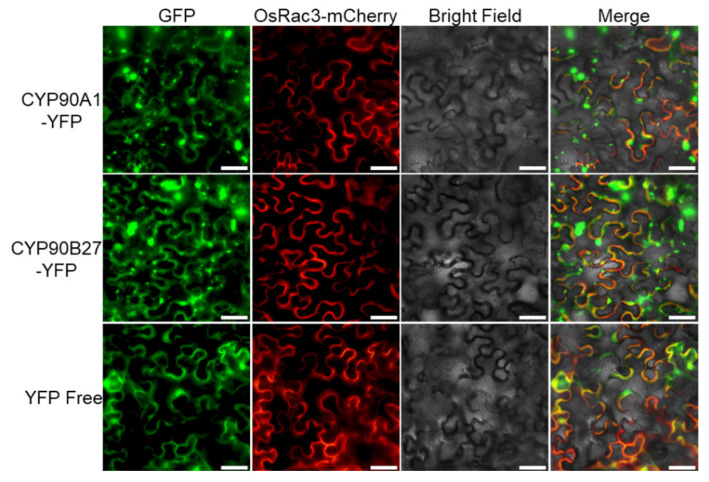
Subcellular localization of CYP90A1 and CYP90B27 in *N. benthamiana* leaves. Transiently expressing p35S::CYP90A1-YFP or p35S::CYP90B27-YFP fusion protein in the leaves of *N. benthamiana* by *A. tumefaciens*. Transient expression of p35S::YFP was used as the control. Subcellular localizations of the fused proteins were analyzed by fluorescence microscopy at 48 h after infiltration. The OsRac3-mCherry fusion protein is a plasma membrane marker. Scale bars = 50 μm.

**Table 1 genes-14-00219-t001:** List of 48 CYP450s of F. cirrhosa identified in this study.

No	Gene ID	Clan	Family	Subfamily	Amino Acid Residues	MW (kDa)	pI	Subcellular Localization
1	27614	71	701	CYP701A6	429	49,317.84	5.73	cytoplasmic
2	25847	71	71	CYP71A1	495	56,263.57	6.78	chloroplast
3	25128	71	71	CYP71A1	496	56,013.99	8.22	chloroplast
4	25333	71	71	CYP71A2	511	56,277.12	6.89	chloroplast
5	24267	71	71	CYP71A25	503	57,395.89	7.69	chloroplast
6	25022	71	71	CYP71A4	498	56,964.15	7.72	vacular membrane
7	19214	71	71	CYP71A9	490	55,311.46	8.42	chloroplast
8	23625	71	71	CYP71D381	495	55,669.95	9.04	chloroplast
9	23436	71	71	CYP71D55	510	58,297.56	9.08	chloroplast
10	27190	71	71	CYP71D8	498	56,365.75	6.48	chloroplast
11	24560	71	73	CYP73A1	505	58,138.45	8.52	plasma membrane
12	22595	71	73	CYP73A100	518	58,650.4	8.6	plasma membrane
13	23153	71	73	CYP73A16	504	57,796.28	9.24	plasma membrane
14	22888	71	75	CYP75B137	503	55,336.92	6.88	chloroplast
15	23199	71	81	CYP81E8	507	56,385.25	7.15	chloroplast
16	24881	71	81	CYP81E8	509	57,543.15	9.06	chloroplast
17	25104	71	81	CYP81F3	494	55,327.71	6.19	chloroplast
18	23883	71	81	CYP81Q32	513	57,720.7	7.26	chloroplast
19	21801	71	84	CYP84A1	515	57,470.57	6.44	chloroplast
20	22019	71	93	CYP93A3	502	56,432.31	5.96	chloroplast
21	25162	71	89	CYP89A2	507	57,250.83	8.82	chloroplast
22	23387	71	89	CYP89A9	498	56,047.58	8.41	chloroplast
23	23217	71	98	CYP98A2	510	57,981.92	8.61	chloroplast
24	14155	72	72	CYP72A14	522	59,146.97	9.19	chloroplast
25	18891	72	72	CYP72A15	516	59,380.97	8.72	cytoplasmic
26	23630	72	72	CYP72A219	518	59,496.39	9.47	chloroplast
27	24173	72	72	CYP72A616	522	59,004.81	9.37	chloroplast
28	25879	72	734	CYP734A6	515	58,157.25	9.44	chloroplast
29	20540	85	707	CYP707A1	468	52,762.44	9.15	chloroplast
30	24431	85	707	CYP707A2	470	53,332.54	8.96	chloroplast
31	23919	85	85	CYP85A1	468	53,353.24	8.65	chloroplast
32	23991	85	90	CYP90A1	510	57,539.32	8.7	cytoplasmic
33	25638	85	90	CYP90B27	483	54,036.79	8.93	chloroplast
34	22207	85	90	CYP90D2	482	54,659.17	8.11	chloroplast
35	24036	86	704	CYP704C1	507	58,078.96	8.42	chloroplast
36	24455	71	77	CYP77A4	509	57,610.25	8.77	chloroplast
37	22541	86	94	CYP94A1	508	57,163.09	9.09	chloroplast
38	27487	86	94	CYP94N1	499	55,985.22	8.47	chloroplast
39	26422	86	94	CYP94B1	489	54,906.75	8.32	chloroplast
40	26493	86	94	CYP94C1	495	55,621.93	6.43	chloroplast
41	23522	86	96	CYP96A15	499	56,957.64	9.08	chloroplast
42	15644	86	97	CYP97A3	614	68,383.46	5.86	chloroplast
43	4877	95	95	CYP95	853	94,883.01	11.48	nucleus
44	26361	710	710	CYP710A11	495	56,010.29	7.19	plasma membrane
45	20822	711	711	CYP711A1	526	58,970.43	9.31	plasma membrane
46	32983	40	40	CYP40	369	40,848.45	5.49	cytoskeleton
47	18441	51	51	CYP51	488	55,580.08	8.13	chloroplast
48	10385	63	63	CYP63	689	76,552.99	10.67	nucleus

## Data Availability

All datasets used in this study are available on FigShare at the link: https://doi.org/10.6084/m9.figshare.21900003.v1.

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
