# Peer review of "Transcriptome Analysis of CYP450 Family Members in Fritillaria cirrhosa D. Don and Profiling of Key CYP450s Related to Isosteroidal Alkaloid Biosynthesis"

_genes, 2023, doi:10.3390/genes14010219_

Round 1
Reviewer 1 Report
This work is on finding and characterizing CYP450 family members on ISA biosynthesis in Fritillaria cirrhosa using both transcriptomics and metabolomics approaches.
Overall, I'm happy with the work but there are some minor corrections that I have highlighted in the pdf file attached to this review. Minor corrections relating to the extraction kits used, the versions of various bioinformatics tools and dates of access of the databases used during the annotation part.
I would also like it if the authors could reorganize the Figures and Tables to be right after the paragraphs that cited those Figures or Tables for easy referencing by the reader.

Author Response
Thanks for your positive comments on this work. We appreciate your useful suggestions, which have significantly improved our manuscript. We have made corrections point to point according to you suggestions, which are shown in the attached file.

Reviewer 2 Report
The minor spelling checks should be made before final submission. The research and presentation has enough worth to be published in this journal.
Author Response
Thanks for your positive comments on this work. We have carefully checked the spellings through the manuscript.
